# Mixed Control Strategy for a Class of Sector-Bounded Nonlinear Systems

**DOI:** 10.3390/e27030261

**Published:** 2025-03-01

**Authors:** Adrian-Mihail Stoica, Isaac Yaesh

**Affiliations:** 1Faculty of Aerospace Engineering, University Politehnica of Bucharest, 060042 Bucharest, Romania; adrian.stoica@upb.ro; 2Control Department, Elbit Systems, Ramat-Hasharon 47100, Israel

**Keywords:** stochastic systems, stochastic antiresonance, sector-bounded nonlinearities, stability analysis, infinitesimal generator, mixed strategies

## Abstract

Here, mixed-strategy-based control of systems with sector-bounded nonlinearities is considered. The suggested control strategy applies a stochastic state feedback, where the control gain includes a white noise component in addition to the deterministic part. While each of the control signal components can sometimes accomplish the control task independently, the combination may have some merits. This is especially true when both the mean value and the variance of the control signal need to be quantified separately. Systems that apply deterministic state-feedback control are abundant, whereas the application of state-multiplicative noise as a mean of control is more limited. Nevertheless, Stochastic Anti Resonance (SAR) with state-multiplicative noise based control, are encountered in diverse engineering applications, physics modelling, and biological models, such as visual-motor tasks. Matrix Inequalities conditions are derived, for weighted $L_{2}$-gain using a mixed strategy control along with exponential $L_{P}$-stability of the closed-loop. A numerical example is given, where the merit of mixed control strategy comparing to deterministic control is demonstrated.

## 1. Introduction

State-feedback controllers with deterministic gains are abundant, whereas the use of stochastic control (i.e., stochastic gains) to achieve stochastic antiresonance (SAR), so as to stabilize unstable nonlinear systems utilizing state-multiplicative noise, is more limited. The reverse case of stochastic resonance (SR), where a noise effect in a nonlinear system evokes resonance or even instability, has been considered in several contexts, such as the periodic occurrences of ice ages in [1]. Additional examples of SR are mentioned in [1,2], in relation to a particle in a double well, animal behavior, biological sensory neurons, ionic channels in cells, optical systems, and electronic devices. The action of SAR squid giant axons has also been considered in [3], where the potential for therapeutic neurological applications was pointed out. Further such SAR applications appear in [4,5,6]. In SR, a bell-shaped signal-to-noise ratio (SNR) as a function of the applied noise intensity is observed, whereas in SAR, the bell shape is upside down, i.e., the noise effect is to reduce the resonance level. When SAR is artificially achieved using a control input that applies state-multiplicative noise to reduce the resonance level, the artificial noise components can be generated using a normally distributed random number generator, appropriately scaled in proportion to the system states. Recently the stabilization of a class of nonlinear sector-bounded nonlinear systems was analyzed, where linear matrix inequalities (LMIs) based conditions have been derived [7]. The present paper is aimed at further developing the SAR idea, by combining stochastic control with deterministic control. To this end, the $L_{2}$-gain of the resulting exponentially $L_{p}$-stable closed-loop system is analyzed using the novel Lyapunov functional of [7], where motivation and some insight into SAR have been provided. To clarify our ideas, we explore the mechanism of SAR and its manifestation in nonlinear systems further. The most abundant example of SAR is a stick balancing on a fingertip. The pioneering work of [8,9] analyzed experimental results and suggested a stochastic inverted pendulum model to reproduce these results. Meanwhile, in [10], further numerical experiments were given, and it was pointed out that the same mechanism, where multiplicative noise has a stabilizing effect, also arises in the context of financial markets. One can gain some insight from the notion of the dynamical traps presented, e.g., in [11]. Loosely speaking, when the stick is placed at an incline, the balancing action is a random zero mean noise that is proportional to the angle the stick forms with the local vertical; therefore, initially large random perturbations of the stick with respect to the vertical are observed. Once the stick crosses the vertical line, the balancing action nulls and the system is trapped in this situation. This is of course, an oversimplified description, and to obtain a more exact description, one can either simulate the system or provide a stability analysis.

The present paper extends the stability results derived in [7] to the so called γ-attenuation problem, which represents a major research topic in control systems theory over the last few decades. There are two main differences with respect to the “classical” γ-attenuation problem (also well known in the deterministic framework as the $H_{\infty }$ norm minimization problem [12]). The first one is the fact that the two-input two-output standard system from the linear deterministic case includes sector-bounded nonlinearities in our formulation. A note in the Conclusion section suggests that based on the universal approximation theorem, the case considered in the present paper may be extended to more general types of nonlinearities. The second particularity consists of the mixed structure of the control law, including both a state-feedback component and a state-multiplicative white noise one. The aim of this hybrid structure is to improve the γ-attenuation-type performance with respect to the case when only a state-feedback deterministic control is used. In order to solve the γ-attenuation problem in the present paper, two main preliminary results are required. Firstly, stability conditions for the considered class of nonlinear stochastic systems, as derived in [7], are briefly presented in the Preliminary section. The second one, is in fact a bounded-real-lemma-type result, providing boundedness conditions for a weighted $L_{2}$-gain associated with the stochastic system with sector-bounded nonlinearities. Its statement and proof are given in Theorem 1 of this paper. Then, combining these results, the mixed control problem including a state-feedback term and a state-dependent white noise component has been formulated and solved in Section 5. The theoretical results are illustrated using a numerical example for a third-order chaos generator model with sector-bounded nonlinearities.

Throughout the paper, $R^{n}$ denotes the *n* dimensional Euclidean space, $R^{n\times m}$ is the set of all n×m real matrices, and the notation X>0 (X≥0) for $X\in R^{n\times n}$ means that *X* is symmetric and positive definite (semi-definite, respectively). Tr{A} denotes the trace of matrix *A* and for $w\in R^{n}$ and ${\left|w\right|}^{2}$ denotes $w^{T}w$. Throughout the paper, (Ω,F,P) is a given probability space. Expectation is denoted by E{·}, and $L^{2}\left(\left[0,\infty \right),R^{m}\right)$ represents the space of all *m*-dimensional measurable stochastic processes f(t) with $E\left\{\int_{0}^{\infty }{\left|f\left(t\right)\right|}^{2}dt\right\}<\infty$.

## 2. Preliminaries

In this section and in the beginning of the next section, we repeat a few preliminary definitions and results regarding the stability of stochastic systems that were already given in [7]. We did this for the sake of self completeness of the paper.Our focus is on stochastic systems with state multiplicative noise that satisfy the following Itô-type stochastic differential equations (SDEs):(1)$$\begin{array}{c} dx(t)=f(x(t),t)dt+g(x(t),t)d\beta (t), \end{array}$$
where β(t) is a zero mean *r*-dimensional Wiener process adapted to an increasing family of $F_{t\geq 0}$ of σ-algebras $F_{t}\subset F$, with $E\{d\beta \left(t\right)d\beta^{T}\left(t\right)\}=Qdt$ and Q≥0. The state vector is $x\left(t\right)\in R^{n}$, and it is assumed that functions f(x(t),t) and g(x(t),t) satisfy the existence conditions for a unique solution of the above stochastic differential equation (see e.g., [13,14,15]). For an initial condition $x_{0}$ at $t=t_{0}$ independent of the σ-algebra generated by β(t),t≥0, this solution will be denoted by $x(t,t_{0},x_{0})$. Assume that f(0,t)=0 and g(0,t)=0, ∀t≥0. Then, according to [16], the trivial solution x(t)≡0 of (Equation 1) is called stable in probability for t≥0 if for any $t_{0}\geq 0$ and ϵ≥0,
$$\begin{array}{c} \lim_{x_{0}\to 0}P\left\{\underset{t\geq t_{0}}{\sup}\left|x\left(t,t_{0},x_{0}\right)\right|>\epsilon \right\}=0. \end{array}$$
Moreover, the solution x(t)≡0 is called asymptotically stable in probability if it is stable in probability and if$$\begin{array}{c} \lim_{x_{0}\to 0}P\left\{\lim_{t\to \infty }x\left(t,t_{0},x_{0}\right)=0\right\}=1. \end{array}$$

One can prove (see, e.g., [16]) that x(t)≡0 is asymptotically stable in probability if there exists a twice continuously differentiable positive definite function V(x,t), such that LV<0, where the infinitesimal generator LV has the expression [14,15](2)$$\begin{array}{c} LV:=V_{t}+V_{x}^{T}f\left(x\left(t\right),t\right)+\frac{1}{2}Tr\left\{g\left(x\left(t\right),t\right)Qg^{T}\left(x\left(t\right),t\right)V_{xx}\right\}, \end{array}$$
in which $V_{t}$ and $V_{x}$ denote the first-order partial derivatives of V(x,t) with respect to *t* and *x*, respectively, and $V_{xx}$ is its second partial derivative with respect to *x*. This result represents a generalization of the well-known Lyapunov’s theorem on asymptotic stability from the deterministic framework. Although this type of a stability is weaker than mean square exponential stability (see, e.g., [17]), it is still of practical value as we will see in the sequel.

## 3. Problem Formulation

Consider the following system:
(3)$$\begin{array}{c} \begin{array}{ccc} dx(t) & = & \left(Ax(t)+Bw(t))dt+Ff(y(t))dt+Dx(t)d\beta (t\right)\\ y\left(t\right) & = & Cx\left(t\right)\\ z\left(t\right) & = & Lx(t)\\ x(0) & = & x_{0} \end{array} \end{array}$$
where $x\in R^{n}$ is the state vector, $y\in R^{n}$ denotes the measured system output, $w\in R^{m}$ is an exogenous input, and $\beta \left(t\right)\in R$ is a standard Wiener process with $E\{\beta^{2}\left(t\right)\}=dt$ on the given probability space, which is also independent of $x_{0}$. The elements of *y* are $y_{i}=C_{i}x\in R,\;i=1,\ldots ,n$ and the components $f_{i}\left(y_{i}\right)$ of f(y) satisfy the sector conditions $0\leq y_{i}f_{i}\left(y_{i}\right)\leq s_{i}y_{i}^{2}$ [18,19,20], which are equivalent to(4)$$f_{i}\left(y_{i}\right)\left(f_{i}\left(y_{i}\right)-s_{i}y_{i}\right)\leq 0,\;i=1,\ldots ,n.$$
In the present paper, we analyze a mixed control procedure for the stabilization of (Equation 3) using a state-feedback control law u(t)=Kx(t) together with the tuning of the intensity *D* of the state dependent noise. Moreover, it will be shown that for a given γ>0, the mixed control strategy may accomplish the following weighted $L_{2}$ gain boundedness condition, expressed as(5)$$J=E\left\{\int_{0}^{\infty }{\left(x^{T}x\right)}^{-\rho }\left[z^{T}\left(t\right)z\left(t\right)-\gamma^{2}w^{T}\left(t\right)w\left(t\right)\right]dt\right\}<0,x_{0}=0$$
for all w(t) satisfying $E\left\{\int_{0}^{\infty }{\left(x^{T}x\right)}^{-\rho }w^{T}\left(t\right)w\left(t\right)dt\right\}<\infty$ for a certain ρ∈(0.5,1).

## 4. Weighted $L_{2}$ Gain Characterization

Assume that the following conditions from [7] are accomplished.

**H1.** 
*The derivatives of the nonlinearities are bounded, namely there exist $\delta_{i}>0$, i=1,…,n, such that $\frac{df_{i}\left(y_{i}\right)}{dy_{i}}<\delta_{i}$, and*


**H2.** 
*The matrix C satisfies the condition $C^{T}C=I$.*


**Remark 1.** 
*The assumption $C^{T}C=I$ may be fulfilled if the matrix $C^{T}C$ is nonsingular when performing a similarity transformation $\hat{x}=\hat{T}x$. See [7] for the details. Note that the case of singular $C^{T}C$ can be readilty aleviated by adding fictitious auxiliary outputs, the effect of which may be nulled by setting $s_{i}=0$ for their corresponding indices i.*


For the sequel, let us define $S=diag\{s_{1},\ldots ,s_{n}\}$ and $\Delta =diag\{\delta_{1},\ldots ,\delta_{n}\}$.

Then, the following result characterizes the conditions for an upper bound on the induced weighted $L_{2}$-norm of the system (Equation 3), for p=ν∈(0,1).

**Theorem 1.** 
*Assume that assumptions H1 and H2 hold. If there exist ν∈(0,1), $\Lambda =diag(\lambda_{1},\ldots ,\lambda_{n})$, $\lambda_{i}\geq 0$, i=1,…,n and $T=diag\left(\tau_{1},\ldots ,\tau_{n}\right),\;\tau_{i}\geq 0$, i=1,…,n, such that*

(6)
$$\begin{array}{c} \left[\begin{array}{ccc} M_{11} & M_{12} & M_{13}\\ {M_{12}}^{T} & M_{22} & M_{22}\\ {M_{13}}^{T} & M_{23}^{T} & M_{33} \end{array}\right]<0, \end{array}$$

*where the following notations have been introduced*

(7)
$$\begin{array}{c} \begin{array}{c} M_{11}:=\frac{\nu }{2}\left(A^{T}+A\right)+\frac{\nu \sigma^{2}}{2}\left(1-2\rho \right)I+\frac{\sigma^{2}}{2}C^{T}\Lambda \Delta C+L^{T}L\\ M_{12}:=\frac{\nu }{2}F+\frac{1}{2}A^{T}C^{T}\Lambda -\frac{\sigma^{2}\rho }{4}C^{T}\Lambda +\frac{1}{2}SC^{T}T\\ M_{22}:=-T+\frac{1}{2}\left(\Lambda CF+F^{T}C^{T}\Lambda \right)\\ M_{13}:=\frac{\nu }{2}B\\ M_{23}:=\frac{1}{2}\Lambda CB\\ M_{33}:=-\gamma^{2}I, \end{array} \end{array}$$

*and $\rho :=1-\frac{\nu }{2}$; then, the solution x(t)≡0 of the stochastic system (Equation 3) with D=σI is asymptotically stable in probability for any sector-type nonlinearities $f_{i}\left(y_{i}\right)$, for which $0\leq y_{i}f_{i}\left(y_{i}\right)\leq s_{i}y_{i}^{2}$ and $\frac{df_{i}\left(y_{i}\right)}{dy_{i}}<\delta_{i}$, i=1,…,n. Moreover, this solution satisfies the weighted $L_{2}$-boundedness condition of (Equation 5).*


**Proof of Theorem 1.** Consider the positive definite function$$\begin{array}{c} V\left(x\right)={\left(x^{T}x\right)}^{\nu /2}+\Sigma_{k=1}^{n}\lambda_{k}\int_{0}^{y_{k}}s^{-2\rho }f_{k}\left(s\right)ds, \end{array}$$
with $\lambda_{k}\geq 0,\;k=1,\ldots ,n$, ν∈(0,1). Then direct computations give that$$\begin{array}{c} V_{x}\left(x\right)=\nu {\left(x^{T}x\right)}^{-\rho }x+{\left(x^{T}C^{T}Cx\right)}^{-\rho }C^{T}\Lambda f \end{array}$$
and$$\begin{array}{c} \begin{array}{ccc} V_{xx}\left(x\right) & = & \frac{\nu }{{\left(x^{T}x\right)}^{\rho }}\left(I-\frac{2\rho }{x^{T}x}xx^{T}\right)-\rho {\left(x^{T}C^{T}Cx\right)}^{-\rho -1}C^{T}\Lambda fx^{T}C^{T}C\\ & & +{\left(x^{T}C^{T}Cx\right)}^{-\rho }C^{T}\Lambda f_{y}C, \end{array} \end{array}$$
where Λ was defined in the statement, and we denote(8)$$\begin{array}{c} \begin{array}{c} f:={\left[f_{1},\ldots ,f_{n}\right]}^{T}\;\mathrm{and}\;f_{y}:=diag\left(\frac{df_{1}}{dy_{1}},\ldots ,\frac{df_{n}}{dy_{n}}\right). \end{array} \end{array}$$
Further, define(9)$$\begin{array}{c} \begin{array}{ccc} -F_{0} & := & LV+\left(z^{T}z-\gamma^{2}w^{T}w\right){\left(x^{T}x\right)}^{-\rho }\\ & = & V_{x}^{T}\left(Ax+Ff+Bw\right)+\frac{1}{2}x^{T}D^{T}V_{xx}Dx\\ & & +\left(z^{T}z-\gamma^{2}w^{T}w\right){\left(x^{T}x\right)}^{-\rho } \end{array} \end{array}$$
and the nonlinearity constraints(10)$$\begin{array}{c} -F_{i}:={\left(x^{T}x\right)}^{-\rho }f_{i}\left(y_{i}\right)\left(f_{i}\left(y_{i}\right)-s_{i}y_{i}\right)\leq 0,\; \end{array}$$
i=1,…,n. Applying the S-procedure technique (see, e.g., [20]), the condition $-F_{0}<0$ is then satisfied together with the constraints (Equation 10) if there exist $\tau_{1},\ldots ,\tau_{n}\geq 0$, such that(11)$$\begin{array}{c} F_{0}-\sum_{i=1}^{n}\tau_{i}F_{i}>0. \end{array}$$Using the above expressions of $V_{x}\left(x\right)$ and $V_{xx}\left(x\right)$, we readily obtain that (Equation 11) is equivalent to(12)$$\begin{array}{c} \begin{array}{c} \left[\nu {\left(x^{T}x\right)}^{-\rho }x^{T}+{\left(x^{T}C^{T}Cx\right)}^{-\rho }f^{T}\Lambda C\right]\left(Ax+Ff+Bw\right)\\ +\frac{1}{2}x^{T}D^{T}\left[\nu {\left(x^{T}x\right)}^{-\rho }\left(I-\frac{2\rho }{x^{T}x}xx^{T}\right)\right.\\ \left.-\rho {\left(x^{T}C^{T}Cx\right)}^{-\rho -1}C^{T}\Lambda fx^{T}C^{T}C\right.\\ \left.+{\left(x^{T}C^{T}Cx\right)}^{-\rho }C^{T}\Lambda f_{y}C\right]Dx\\ -{\left(x^{T}x\right)}^{-\rho }\left(f^{T}Tf-\frac{1}{2}f^{T}TCSx-\frac{1}{2}x^{T}SC^{T}Tf\right)\\ +\left(z^{T}z-\gamma^{2}w^{T}w\right){\left(x^{T}x\right)}^{-\rho }<0 \end{array} \end{array}$$
where T is defined in the Theorem statement.Multiplying (Equation 12) by ${\left(x^{T}x\right)}^{\rho }$, for $C^{T}C=I$, the following is obtained(13)$$\begin{array}{c} \begin{array}{c} \left(\nu x^{T}+f^{T}\Lambda C\right)\left(Ax+Ff+Bw\right)+\frac{1}{2}x^{T}D^{T}\left[\nu \left(I-\frac{2\rho }{x^{T}x}xx^{T}\right)\right.\\ \left.-\frac{\rho }{x^{T}x}C^{T}\Lambda fx^{T}+C^{T}\Lambda f_{y}C\right]Dx-f^{T}Tf+\frac{1}{2}f^{T}TCSx\\ +\frac{1}{2}x^{T}SC^{T}Tf+x^{T}L^{T}Lx-\gamma^{2}w^{T}w<0. \end{array} \end{array}$$Then, for D=σI, the inequality (Equation 13) becomes(14)$$\begin{array}{c} \begin{array}{c} \left(\nu x^{T}+f^{T}\Lambda C\right)\left(Ax+Ff+Bw\right)+\frac{\nu \sigma^{2}}{2}\left(1-2\rho \right)x^{T}x\\ -\frac{1}{2}\sigma^{2}\rho x^{T}C^{T}\Lambda f+\frac{1}{2}\sigma^{2}x^{T}C^{T}\Lambda f_{y}Cx-f^{T}Tf+\frac{1}{2}f^{T}TCSx\\ +\frac{1}{2}x^{T}SC^{T}Tf+x^{T}L^{T}Lx-\gamma^{2}w^{T}w<0, \end{array} \end{array}$$
which may be rewritten as(15)$$\begin{array}{c} \left[\begin{array}{ccc} x^{T} & f^{T} & w^{T} \end{array}\right]\left[\begin{array}{ccc} M_{11} & M_{12} & M_{13}\\ {M_{12}}^{T} & M_{22} & M_{23}\\ {M_{13}}^{T} & M_{23}^{T} & M_{33} \end{array}\right]\left[\begin{array}{c} x\\ f\\ w \end{array}\right]<0, \end{array}$$
with the definitions of (Equation 7). Using the definition of Δ from the statement, it follows that if condition (Equation 6) is accomplished, then the inequality (Equation 15) holds for any ${\left[x^{T}\;\;f^{T}\;\;w^{T}\right]}^{T}\neq 0$. Thus, it can be concluded that(16)$$\begin{array}{c} LV\cdot {\left(x^{T}x\right)}^{\rho }+z^{T}z-\gamma^{2}w^{T}w<0 \end{array}$$
together with the sector constraints $f_{i}\left(y_{i}\right)\left(f_{i}\left(y_{i}\right)-s_{i}y_{i}\right)\leq 0$, i=1,…,n. On the other hand, according to the Itô’s formula (see, e.g., [13]) for the function V(x) with the solution x(t) of the stochastic differential Equation (Equation 3), it follows that$$\begin{array}{c} \begin{array}{ccc} dV(x) & = & V_{x}^{T}\left(Ax+Bw+Ff(y)\right)dt+\frac{1}{2}Tr\left\{x^{T}D^{T}Dx\right\}dt+V_{x}^{T}Dxd\beta \\ & = & LVdt+V_{x}^{T}Dxd\beta \end{array} \end{array}$$
from which, based on (Equation 16), it follows that$$\left(dV-V_{x}^{T}Dxd\beta \right){\left(x^{T}x\right)}^{\rho }+\left(z^{T}z-\gamma^{2}w^{T}w\right)dt<0$$
or, equivalently,$$\left(dV-V_{x}^{T}Dxd\beta \right)+{\left(x^{T}x\right)}^{-\rho }\left(z^{T}z-\gamma^{2}w^{T}w\right)dt<0.$$Integrating the latter equation from 0 to *T*, we obtain$$V\left(x\left(T\right)\right)-V\left(x_{0}\right)-\int_{0}^{T}V_{x}^{T}Dxd\beta +\int_{0}^{T}{\left(x^{T}x\right)}^{-\rho }\left(z^{T}z-\gamma^{2}w^{T}w\right)dt<0.$$On the other hand, from (Equation 6) in the statement of Theorem 1 it follows, by subtracting the non-negative term $L^{T}L$ from the left-upper block, that(17)$$\begin{array}{c} \left[\begin{array}{cc} M_{11}-L^{T}L & M_{12}\\ M_{12}^{T} & M_{22} \end{array}\right]<0. \end{array}$$
But, in [7], it is proved that if condition (Equation 17) holds, then the solution x(t)≡0 is also exponentially *p*-stable for p=ν (see, e.g., [16]) for w=0. Namely, there exists α,β>0, such that $E\left[x\left(t,x_{0}\right){|}^{p}\right]\leq \alpha {\left|x_{0}\right|}^{p}e^{-\beta t}$.Further, taking the limit of T→∞ and noting that $E\{\int_{0}^{\infty }V_{x}^{T}Dxd\beta \}=0$ (see, e.g., [13,15]), we readily obtain that (Equation 5) holds.  □

## 5. Mixed Strategy Control

Given the above bounded-real-lemma like result, we may now consider the mixed-strategy control of(18)$$\begin{array}{c} \begin{array}{ccc} dx(t) & = & \left(\bar{A}x\left(t\right)+B_{1}w\left(t\right)+B_{2}u\left(t\right)\right)dt+Ff\left(y\left(t\right)\right)dt+Dx\left(t\right)d\beta \left(t\right)\\ y\left(t\right) & = & Cx\left(t\right)\\ z\left(t\right) & = & C_{1}x\left(t\right)+D_{12}u\left(t\right)\\ x(0) & = & 0 \end{array} \end{array}$$
where $x\in R^{n}$ is the state vector, $y\in R^{n}$ is the measured system output, *z* is the controlled output, and $u\in R^{m}$ is the deterministic component of the control signal, whereas Dx(t)dβ(t) is the random component, namely the state-multiplicative noise introduced to implement a mixed strategy control signal. Assuming full access to the state-vector *x*, we can write(19)$$\begin{array}{c} u(t)=Kx(t) \end{array}$$
leading to $A=\bar{A}+B_{2}K$. Then, applying Theorem 1 for the resulting system (Equation 18) obtained with the control law (Equation 19) and denoting $R:=D_{12}^{T}D_{12}$, assuming $D_{12}^{T}C_{1}=0$, if follows that condition (Equation 6) may be written in the equivalent form(20)$$\left[\begin{array}{cccc} N_{11} & N_{12} & N_{13} & K^{T}\\ {N_{12}}^{T} & N_{22} & N_{23} & 0\\ {N_{13}}^{T} & N_{23}^{T} & N_{33} & 0\\ K & 0 & 0 & -R^{-1} \end{array}\right]<0,$$
where$$\begin{array}{c} \begin{array}{c} N_{11}:=\frac{\nu }{2}\left({\bar{A}}^{T}+\bar{A}+K^{T}B_{2}^{T}+B_{2}K\right)+\frac{\nu \sigma^{2}}{2}\left(1-2\rho \right)I+\frac{\sigma^{2}}{2}C^{T}\Lambda \Delta C+C_{1}^{T}C_{1}\\ N_{12}:=\frac{\nu }{2}F+\frac{1}{2}\left({\bar{A}}^{T}+K^{T}B_{2}^{T}\right)C^{T}\Lambda -\frac{\sigma^{2}\rho }{4}C^{T}\Lambda +\frac{1}{2}SC^{T}T\\ N_{22}:=-T+\frac{1}{2}\left(\Lambda CF+F^{T}C^{T}\Lambda \right)\\ N_{13}:=\frac{\nu }{2}B_{1}\\ N_{23}:=\frac{1}{2}\Lambda CB_{1}\\ N_{33}:=-\gamma^{2}I. \end{array} \end{array}$$

Note that one may interpret the following(21)$$\begin{array}{c} du_{mixed}\left(t\right):=B_{2}Kx\left(t\right)dt+Dx\left(t\right)d\beta \left(t\right) \end{array}$$
as the overall control signal produced by the mixed strategy controller, with both a drift and diffusion parts [21]. Therefore, the overall control effort is given using the result of Theorem 4.4 in [21] regarding the second-order stochastic integral in $d\beta^{2}$, by(22)$$\begin{array}{c} \begin{array}{ccc} E\left\{\int_{0}^{\infty }{\left|du_{mixed}\right|}^{2}dt\right\} & = & E\left\{\int_{0}^{\infty }x^{T}K^{T}B_{2}B_{2}^{T}Kxdt+x^{T}D^{T}Dxd\beta^{2}\left(t\right)\right\}\\ & = & E\left\{\int_{0}^{\infty }x^{T}\left(K^{T}B_{2}^{T}B_{2}K+D^{T}D\right)xdt\right\}. \end{array} \end{array}$$

However, by fixing D=σI, one can minimize just the effect $E\{\int_{0}^{\infty }x^{T}\left(K^{T}B_{2}^{T}B_{2}K+\sigma^{2}I\right)xdt\}$ of the deterministic control component for different values of $\sigma^{2}$, hence taking it as a design parameter.

The mixed strategy control parameters $K,\sigma^{2}$, are, therefore, determined by solving the matrix inequality (Equation 20) for Λ≥0T≥0, *K* and $\sigma^{2}>0$, where ν∈(0,1) is found using line search.

## 6. A Numerical Example

Consider the third-order system of form (Equation 18) with a single nonlinearity in which(23)$$\begin{array}{c} \bar{A}=\left[\begin{array}{ccc} -\epsilon & 1 & 0\\ 0 & -\epsilon \; & 1\\ a_{1} & a_{2} & a_{3} \end{array}\right],\;B_{1}=B_{2}=\left[\begin{array}{c} 0\\ 0\\ 1 \end{array}\right],\;C=I_{3},\\ F=\left[\begin{array}{ccc} 0 & 0 & 0\\ 0 & 0 & 0\\ 10 & 0 & 0 \end{array}\right],C_{1}=\left[\begin{array}{ccc} 1 & 0 & 0\\ 0 & 0 & 0 \end{array}\right],\;D_{12}=\left[\begin{array}{c} 0\\ r \end{array}\right] \end{array}$$
where $a_{1}=-2,a_{2}=-1.48,a_{3}=-1,\epsilon =0.01$ and r>0. This system is a slightly modified version of the third-order chaos generator model with a single nonlinearity considered in [22]. Further, the feasibility condition (Equation 20) was checked for different values of $R=r^{2}$, with the aim of minimizing the level of attenuation γ. The numerical results obtained for three different values of *R* are presented in Table 1, together with the corresponding values of σ, $Tr\{K^{T}K\}$, and of the overall control effort defined in (Equation 22), where the third line in the table represents the (almost) purely stochastic control case.

In Figure 1, the time responses of the three states and of the overall control effort corresponding to R=1 are presented. These plots are generated applying a null control (u(t)≡0) for t∈[0,100) s and the mixed strategy control of form (Equation 21) for t∈[100,200] s. $\alpha (A+B_{2}K)$ given in the caption of Figure 1 is defined as the largest real part of the eigenvalues of $A+B_{2}K$. It can be seen that, although the matrix $A+B_{2}K$ is not Hurwitz, the system is stabilized due to the state-multiplicative noise component of the control. For comparison, Figure 2 depicts the (almost) purely stochastic control case.

## 7. Discussion

At this point, one may observe the lack of insight regarding the weighted induced norm relation of (Equation 5). To gain some insight, consider the above derivations, where$$LV+{\left(x^{T}x\right)}^{-\rho }\left(z^{T}z-\gamma^{2}w^{T}w\right)=\left[\begin{array}{cc} \xi^{T} & w^{T} \end{array}\right]\tilde{M}\left[\begin{array}{c} \xi \\ w \end{array}\right]$$
where $\xi :=\left[\begin{array}{c} x\\ f \end{array}\right]$ and where$$\tilde{M}:=\left[\begin{array}{cc} {\tilde{M}}_{11} & {\tilde{M}}_{12}\\ {\tilde{M}}_{12}^{T} & {\tilde{M}}_{22} \end{array}\right].$$
Here,$${\tilde{M}}_{11}:=\left[\begin{array}{cc} M_{11} & M_{12}\\ M_{12}^{T} & M_{22} \end{array}\right],{\tilde{M}}_{12}:=\left[\begin{array}{c} M_{13}\\ M_{23} \end{array}\right],{\tilde{M}}_{22}=M_{33}.$$
Completing to squares we, readily, obtain that$$\begin{array}{c} \begin{array}{c} LV+{\left(x^{T}x\right)}^{-\rho }\left(z^{T}z-\gamma^{2}w^{T}w\right)=\left(w^{T}+\xi^{T}{\tilde{M}}_{12}{\tilde{M}}_{22}^{-1}\right){\tilde{M}}_{22}\left(w+{\tilde{M}}_{22}^{-1}{\tilde{M}}_{12}^{T}\xi \right)\\ +\xi^{T}\left({\tilde{M}}_{11}-{\tilde{M}}_{12}{\tilde{M}}_{22}^{-1}{\tilde{M}}_{12}\right)\xi . \end{array} \end{array}$$
As $\tilde{M}<0$, the RHS of the latter is maximized by $w=-{\tilde{M}}_{22}^{-1}{\tilde{M}}_{12}^{T}\xi$, where we recall that $\xi :=\left[\begin{array}{c} x\\ f \end{array}\right]$. Keeping in mind that z=Lx and the sector-bounds (Equation 4) on $f_{i}$ ensures $|f_{i}|<s_{i}C_{i}x$, we see that the maximal value of $LV+{\left(x^{T}x\right)}^{-\rho }\left(z^{T}z-\gamma^{2}w^{T}w\right)$ is of the order $x^{T}x/{\left(x^{T}x\right)}^{\rho }={\left(x^{T}x\right)}^{\nu /2}$. Namely, loosely speaking, the weighted $L_{2}$-norm relation of (Equation 5) is closely related to the non-weighted $L_{p}$-norm, with p=ν/2. In the limiting case of ν→0, it is actually related to the $L_{0}$ norm, which, in its discrete version of $\ell_{0}$ norm, measures the number of non-zero elements of x(t). This may provide some explanation regarding the simulations in the previous numerical example, where SAR-based control using state-multiplicative noise nulls *z* most of the time.

## 8. Conclusions

The phenomenon of stochastic antiresonance (SAR) for a class of systems with sector-bounded nonlinearities has been utilized for the synthesis of a mixed strategy control that consists of a deterministic state-feedback and state-multiplicative noise serving as the random part of the control signal. The control design involves bilinear matrix inequalities, the solutions of which are supported by YALMIP [23], for example by using the nonconvex quadratic solver *quadprogbb*. The merits of the mixed strategy over the deterministic state feedback of state-multiplicative control noise are demonstrated using a simple example.The considered class of systems with sector-bounded nonlinearities correspond to a large number of practical applications. It’s applications can be expanded through the universal approximation theorem [24] to more general systems that have not a priori given in terms of sector-bounded nonlinearities. Such systems may be approximated with an arbitrarily small error, by neural networks with a single hidden layer, with, e.g., a tanh activation function and a linear output layer. One such example is the Morris–Lecar model [25] of a neuron. The application of SAR in this model and other similar ones will be considered in future research.

## Figures and Tables

**Figure 1 entropy-27-00261-f001:**
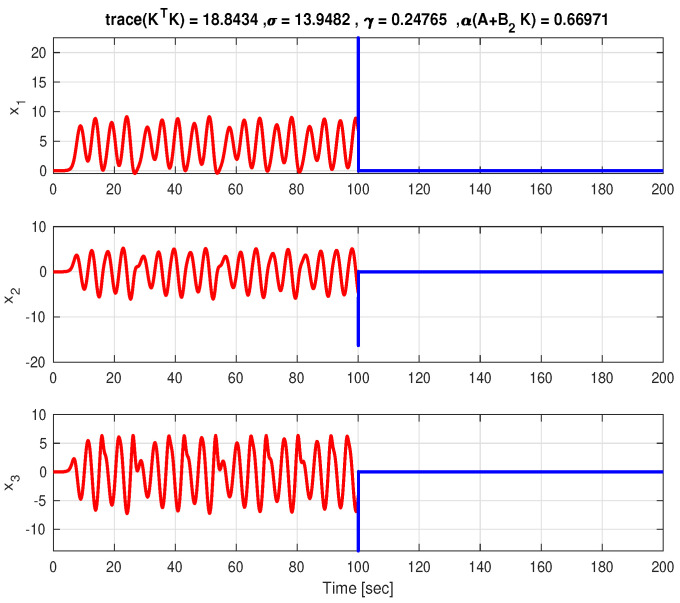
The time responses of the states for the case R=1.

**Figure 2 entropy-27-00261-f002:**
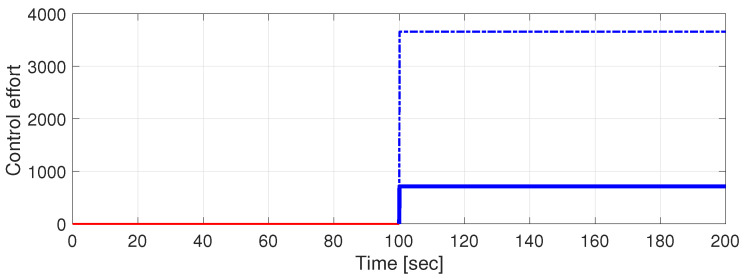
The time responses of the states and of the overall control effort for R=1 (dotted) and for the case $R=10^{6}$ (solid).

**Table 1 entropy-27-00261-t001:** Numerical Results.

*R*	minγ	σ	$Tr\{K^{T}K\}$	$E\left\{\int_{0}^{\infty }{\left|u_{mixed}\right|}^{2}dt\right\}$
1	0.2476	13.9482	18.8434	$3.3680\times 10^{3}$
$10^{-6}$	0.1857	28.2044	$5.4252\times 10^{5}$	$2.8471\times 10^{3}$
$10^{6}$	0.0236	23.2726	$1.7262\times 10^{-7}$	714.6595

## Data Availability

No new data were created or analyzed in this study. Data sharing is not applicable to this article.

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
