# Peer review of "Mixed Control Strategy for a Class of Sector-Bounded Nonlinear Systems"

_entropy, 2025, doi:10.3390/e27030261_

Round 1

Reviewer 1 Report

Comments and Suggestions for Authors

This paper proposes an innovative mixed control strategy and makes contributions at both theoretical and numerical levels. 
However, improvements are needed in terms of theory and application. 
The authors are advised to address the issues below to enhance the academic value and engineering significance of the paper.

1. Further clarification is needed on the physical meaning of the stochastic noise in the mixed strategy and its feasibility in
practical systems, e.g., how to generate the state-multiplicative noise.

2. The comparative analysis between SAR and Stochastic Resonance (SR) is insufficient. It is recommended to supplement 
the discussion on their applicability differences in various scenarios.

3. From (19), $u(t)=Kx(t)$. Then $du(t)=Kdx(t)$. How can we interpret the control signal generated by the mixed strategy 
controller as (21)?

4. The comparison is only made with purely deterministic control, and not with purely stochastic control 
(only state-multiplicative noise), making it difficult to fully assess the superiority of the proposed strategy.

5. It is recommended to supplement simulations or experiments on actual physical systems,
to validate the engineering applicability of the method.

Author Response

This paper proposes an innovative mixed control strategy and makes contributions at both theoretical and numerical levels.

However, improvements are needed in terms of theory and application.

The authors are advised to address the issues below to enhance the academic value and engineering significance of the paper.

Comments 1: Further clarification is needed on the physical meaning of the stochastic noise in the mixed strategy and its feasibility in practical systems, e.g., how to generate the state-multiplicative noise.

Response 1: More explanations are given now in the introduction section, to better explain the physical meaning of SAR and new references [14] and [15] are given to support those explanations. Also the generation of state-multiplicative noise is mentioned.

Comments 2: The comparative analysis between SAR and Stochastic Resonance (SR) is insufficient. It is recommended to supplement the discussion on their applicability differences in various scenarios.

  Response 2: The reverse (in terms of SNR) properties of SR and SAR are also briefly discussed now in the introduction section

Comments 3: From (19), $u(t)=Kx(t)$. Then $du(t)=Kdx(t)$. How can we interpret the control signal generated by the mixed strategy controller as (21)?

Response 3: Formulas (21) and (22) have been corrected. Thank you for your comment.

 Comments 4: The comparison is only made with purely deterministic control, and not with purely stochastic control (only state-multiplicative noise), making it difficult to fully assess the superiority of the proposed strategy

Response 4: New simulation for the third line in Table which is nearly purely stochastic, is now simulated in Section 6.

Comments 5: It is recommended to supplement simulations or experiments on actual physical systems,to validate the engineering applicability of the method.

Response 5: A more realistic example of the Morris-Leccar neuron is now mentioned in the conclusions section. Two new references [23],[24] are now cited in this context.

We thank the Reviewer for the time and effort spent in reviewing our paper and for the useful suggestions.

Reviewer 2 Report

Comments and Suggestions for Authors

The introduction must be completely rewritten so that readers understand the main theme of this paper. What is stochastic antiresonance? Why is state-multiplicative noise utilized to stabilize systems? Although the authors mentioned some examples, the readers in the control area are unfamiliar with them. The introduction is too short to emphasize the motivation and contributions of this paper.

It is not clear what you mean by the mixed control strategy. In l. 135, “one may interpret” needs to be rewritten more clearly. (21) must be rewritten more in detail. How do you relate (19) with (21)?

The structure of the introduction is not well organized.

It is not clear why the authors consider the stochastic differential equation.

In addition, this paper's contribution to control theory is unclear.

A more detailed derivation for (17) must be shown.

Author Response

Comments 1: The introduction must be completely rewritten so that readers understand the main theme of this paper. What is stochastic antiresonance? Why is state-multiplicative noise utilized to stabilize systems? Although the authors mentioned some examples, the readers in the control area are unfamiliar with them. The introduction is too short to emphasize the motivation and contributions of this paper.

Response 1: We added more information related to SAR and added an additional relevant application. More explanations are given now in the introduction section, to better explain the physical meaning of SAR and new references [14] and [15] are given to support those explanations. Also the generation of state-multiplicative noise is mentioned.  A more realistic example of the Morris-Leccar neuron is now mentioned in the conclusions section. Two new references [23],[24] are now cited in this context.

Comments 2:  It is not clear what you mean by the mixed control strategy. In l. 135, “one may interpret” needs to be rewritten more clearly. (21) must be rewritten more in detail. How do you relate (19) with (21)?

Response 2: Indeed we had a mistake in (21) and we corrected (21) and (22) accordingly. We hope it is clear now. Thank you.

Comments 3: The structure of the introduction is not well organized.

Response 3: We hope that the new explanations we added improved the readability of the introduction. If it is still not clear or not properly organized, we would be grateful for specific suggestions.

Comments 4: It is not clear why the authors consider the stochastic differential equation.

Response 4: An explanation about the role of the stochastic differential equation, is now given in the end of Section 1 – the Introduction section.

Comments 5:  A more detailed derivation for (17) must be shown.

Response 5: The reasoning for (17) is now explained, just before the equation.

We thank the Reviewer for the time and effort spent in reviewing our paper and for the useful suggestions.

Round 2

Reviewer 1 Report

Comments and Suggestions for Authors

The authors answered the questions well. I have no further comment.

Author Response

Dear Editor-in-Chief

Thank you very much for your suggestions regarding our manuscript entitled “Mixed Control Strategy for a Class of Sector Bounded Non Linear Systems”, ID: entropy-3467949, submitted for publication at MDPI-Entropy Journal.

We inform you that based based on your suggestions, we introduced a new paragraph in the Introduction (marked with brown color) and a new reference ([16]) required by this new paragraph.
We hope that the new version clarifies the differences with respect to the results derived in [13] and clearly presents the new developments of the present paper.

Sincerely yours,

Isaac Yaesh and Adrian-Mihail Stoica

Reviewer 2 Report

Comments and Suggestions for Authors

no further comments.

Author Response

(The authors gave the same response as above.)
